# Ni-Based Catalyst Derived from NiAl Layered Double Hydroxide for Vapor Phase Catalytic Exchange between Hydrogen and Water

**DOI:** 10.3390/nano9121688

**Published:** 2019-11-25

**Authors:** Xiaoyu Hu, Peilong Li, Xin Zhang, Bin Yu, Chao Lv, Ning Zeng, Junhong Luo, Zhi Zhang, Jiangfeng Song, Yong Liu

**Affiliations:** 1School of Resource Environmental and Safety Engineering, University of South China, Hengyang 421001, China; hu822066398@163.com; 2Institute of Materials, China Academy of Engineering Physics, Jiangyou 621908, China; lipeilong2012@126.com (P.L.); a753951_xin@163.com (X.Z.); yubincaep@163.com (B.Y.); lvchao219@foxmail.com (C.L.); 15882895785@139.com (N.Z.); luojunhong@caep.cn (J.L.); zhangzhi@caep.cn (Z.Z.)

**Keywords:** vapor phase catalytic exchange, layered double oxides, Ni catalyst, water detritiation

## Abstract

A high-efficient and low-cost catalyst on hydrogen isotope separation between hydrogen and water is an essential factor in industrial application for heavy water production and water detritiation. In past studies, Pt-based catalysts were developed but not practical for commercial use due to their high cost for vapor phase catalytic exchange (VPCE), while for impregnated nickel catalysts with a lower cost the problems of agglomeration and low Ni utilization existed. Therefore, to solve these problems, in-situ grown Ni-based catalysts (NiAl-LDO) derived from a layered double hydroxide (LDH) precursor were fabricated and first applied in VPCE in this work. Compared with traditional impregnated Ni-based catalysts, NiAl-LDO catalysts own a unique layered structure, homogeneous dispersed metallic phase, higher specific surface area as well as stronger metal-support interactions to prevent active metal from agglomerating. These advantages are beneficial for exposing more active sites to improve dynamic contacts between H_2_ and HDO in a catalyst surface and can bring excellent catalytic activity under a reaction temperature of lower than 400 °C. Additionally, we found that the dissociative chemisorption of HDO and H_2_ occurs not only in Ni (111) but also in NiO species where chemisorbed H(ads), D(ads), OH(ads) and OD(ads) are formed. The results highlight that both of the Ni^2+^ species and Ni^0^ species possess catalytic activities for VPCE process.

## 1. Introduction

Tritium plays a pivotal role in nuclear power plants, nuclear fuel reprocessing, as well as future fusion reactors [1,2]. However, tritiated water damages the human body and environment seriously due to its radioactivity and toxicity. Thus, stringent environment standards for the development of inland nuclear power were established, like in China. Besides, the treatment of tritiated water possesses a great significance in nuclear energy and environment [3,4]. Hydrogen isotope exchange is an important process for tritium extraction from heavy water and detritiation of tritium-containing sewage [5,6,7]. Particularly, the vapor phase catalytic exchange (VPCE) reaction, which processes the tritium, is transferred from vaporous water into gaseous hydrogen (HTO_vapor_ + H_2gas_ ⇌ HT_gas_ + H_2_O_gas_) and has attracted attention due to its nontoxicity, non-corrosiveness, simple craft as well as a high separation factor throughout the hydrogen isotope exchange reaction. Hence is owes a huge potential for industrial application [8,9,10,11,12,13]. In VPCE, the availability of active sites (transport of gaseous hydrogen and vaporous water to a catalyst surface as well as the dissociative chemisorption) is key for the hydrogen isotope exchange reaction. To achieve a highly efficient catalytic exchange reaction, some strategies have been developed such as enhancing the dispersion and utilization of active metals (Ni, Cr, Pd, Pt, Rh, Ru etc.), using high specific surface area supports as well as the cooperation of multiple metals [10,12,14]. Meanwhile, a mass of hydrophobic catalysts have been fabricated including Pt/C, Pt–Ir/C, Pt–Fe/C, Pt–Ni/C [10,11,13,15]. The impregnated Pt/Al_2_O_3_ and Ni/Al_2_O_3_ [16,17,18] are two kinds of widely researched catalysts for VPCE. Pt/Al_2_O_3_ catalysts have excellent activity, but their extravagant cost is not profitable for commercial application. Traditional impregnated Ni/Al_2_O_3_ catalysts are not able to be used efficiently due to the inevitable aggregation of active metal after calcination and reduction. Accordingly, it is highly desirable to develop low-cost catalysts, like Ni-based catalysts, to realize high catalytic performance for VPCE. The substitutes of Pt-based catalysts are due to their similar property for dissociative chemisorption of reactants on the Ni-based catalysts’ surface. Unfortunately, most Ni-based catalysts were prepared by impregnating metal precursors on supports, followed by calcining and reducing, resulting in the unexpected uncontrollable aggregation of metal species and inefficient availability of active metal. For instance, Zhang et al. prepared Al_2_O_3_ supported by Ni catalysts using the impregnation method, but extra Ni loading caused serious Ni aggregation, blocking active sites and leading to poor CO_2_ methanation performance [19]. To improve metal dispersion and catalytic performances, some researchers prepared in-situ catalysts for hydrodechlorination [20] and methane decomposition [21]. Therefore, it is very significant to explore an in-situ grown Ni catalyst with an outstanding dispersion to prevent active metal from aggregating in supports for VPCE process.

Layered double hydroxide (LDH), also known as hydrotalcite-like materials or anionic clays, are a significant class of layered-structure materials with a chemical composition expressed by the general formula [$M_{1-x}^{2+}$$M_{x}^{3+}$(OH)_2_]^x+^[$A_{x/n}^{n-}$·yH_2_O]^x−^ where M^2+^ and M^3+^ are di- and trivalent metal cations, respectively, and A^n−^ denotes an organic or inorganic anion with a n-valent anion. The unique supramolecular structure with a uniform dispersion for M^2+^ and M^3+^ ions within the layers is from precise control of both the inorganic layers and the interlayer gallery anions of LDH without the formation of “lakes” cations [22]. The adjustability of the hydrotalcite structure gives it uniquely adjustable properties, which are further applicable in numerous areas. Various literature reported the wide applications of LDH as catalyst precursors, catalyst supports, adsorbents and ion exchangers in different research directions, e.g., gas removing, selective hydrogenation, electro-catalysis oxidation etc. [23,24,25]. Especially, when using LDH as catalyst precursors, the LDH-derived catalyst could maintain a layer structure and keep a highly dispersed metallic phase [26]. Importantly, some active metal (Ni, Cr, Pd, Pt etc.) for hydrogen isotope exchange processes could be easily introduced into the LDH by using a hydrothermal method and co-precipitation, ion-exchange, precipitation-reduction method [23]. These advantages of LDH inspire us to fabricate an efficient VPCE catalyst from LDH.

In this work, we report the effective in-situ growth of NiAl layered double oxides (LDO) derived from the LDH precursor which are firstly applied for the VPCE process. The layered structure, high specific surface area, porous structure, homogeneous metallic phase as well as stronger metal-support interactions of in-situ Ni catalysts contribute to the much higher VPCE activity in comparison with the impregnated NiO/γ-Al_2_O_3_ catalysts at a reaction temperature of below 400 °C. Importantly, the catalytic exchange mechanism of transition-metal Ni for VPCE is also discussed, both Ni^0^ species and NiO species can conduce hydrogen isotope exchange processes between hydrogen and water.

## 2. Experimental

### 2.1. Material

Nickel nitrate hexahydrate (Ni(NO_3_)_2_·6H_2_O, 98%), aluminum nitrate nonahydrate (Al(NO_3_)_3_·9H_2_O, 98%) and urea ((NH_2_)_2_CO, 98%) were obtained from Aladdin Chemical Reagent Co. Ltd. (Shanghai, China). Activated alumina was purchased from Macklin Chemical Reagent Co. Ltd. (Shanghai, China). All chemicals were analytical grade and used without further purification. The used ultrapure water throughout the experimental processes was obtained from a Milli-Q ultrapure system (10.5 MΩ cm).

### 2.2. Synthesis of Petal-Like NiAl-LDH

The NiAl-LDH was synthesized by the conventional urea hydrolysis method [25,27,28]. In a typical procedure, as illustrated in Scheme 1A, Ni(NO_3_)_2_·6H_2_O (0.012 mol) and Al(NO_3_)_3_·9H_2_O (0.006 mol) as well as urea (0.084 mol) with a Ni/Al/urea molar ratio of 2/1/14 were dissolved in 320 mL ultrapure water to form a mixed solution with vigorous stirring. Subsequently, the homogeneous and transparent solution was transferred into 500 mL Teflon-lined stainless-steel autoclave, sealed and heated with oven at 150 °C for 12 h. After cooling the autoclave to 25 °C, the precipitate was collected and thoroughly washed by ultrapure water several times, followed by drying at 70 °C overnight in an oven to get NiAl-LDH.

### 2.3. Preparation of Spherical Ni(NO_3_)_2_/γ-Al_2_O_3_

Ni(NO_3_)_2_/γ-Al_2_O_3_ was prepared by a traditional impregnation method [20]. Typically, as illustrated in Scheme 1B, 10 g γ-Al_2_O_3_ was placed in a beaker loaded with 12.5 mL 3 mol L^−1^ Ni(NO_3_)_2_ water solution and then impregnated for 12 h at room temperature, followed by drying at 70 °C for 12 h. The products were named Ni(NO_3_)_2_/γ-Al_2_O_3_ and collected in the bottle to wait for further treating.

### 2.4. Preparations of NiAl-LDO and NiO/γ-Al_2_O_3_

The precursors NiAl-LDH and Ni(NO_3_)_2_/γ-Al_2_O_3_ were calcinated at 500 °C for 4 h with a heating rate of 5 °C min^−1^ in air. After slowly cooling to 25 °C, the products were named as NiAl-LDO and NiO/γ-Al_2_O_3_, respectively.

### 2.5. Characterization

Powder X-ray diffraction (XRD) patterns were recorded using a DX-2700 X-ray powder diffractometer (Haoyuan Inc., Liaoning, China) with a Cu Ka radiation source (λ = 1.54059 Å) in 2 range between 5° and 80°, step size 0.05°, and operated at a voltage of 40 kV and a current of 30 mA. Inductively coupled plasma optical emission spectrometer (ICP-OES) (Agilent, 725ES) (Palo Alto, CA, USA) was applied to identify the elemental content of Ni and Al. The morphologies of samples were performed using a scanning electron microscope (SEM, FEI Quanta 650) (Hillsboro, OR, USA). The specimen positioned on a mesh grid was examined using transmission electron microscopy coupled to the energy-dispersive X-ray spectroscopy (TEM, FEI Talos F200S) (Hillsboro, OR, USA) at a voltage of 200 kV. Selected area electron diffraction (SAED) was performed in TEM (FEI Talos F200S) and TEM with energy dispersive X-ray spectroscopy mapping (TEM EDS maps) (FEI Talos F200S) was also required to recognize the distributions of Ni, Al, C and O for NiAl-LDH and NiAl-LDO. Low-temperature N_2_ adsorption-desorption experiments were conducted using a Micromeritics ASAP 2469 instrument (Norcross, GA, USA). Specific surface area and average pore diameters were calculated by the Brunauer-Emmett-Teller (BET) method or the Barrett-Joyner-Halenda (BJH) method applied to desorption isotherms. The surface valence states of the prepared samples before and after reaction were analyzed by X-ray photoelectron spectroscopy (XPS) (ESCALAB, 250Xi) (Waltham, MA, USA).

H_2_ temperature-programmed reduction (H_2_-TPR) of the samples were conducted on a chemical adsorption apparatus (AMI-300) (Bayport, TX, USA) equipped with a thermal conductivity detector (TCD) signal detector. Prior to the H_2_-TPR experiment, about 0.1000 g of NiAl-LDH powder (Ni 36.75 wt.%, ICP) was loaded in a quartz reactor and were further pre-treated with continuous Ar flow at 70 °C for 30 min, followed by cooling down to 30 °C. Then, the H_2_-TPR of the NiAl-LDH was performed with a ramp rate of 5 °C min^−1^ in a stream of 15 cc min^−1^ 10% H_2_ in Ar from 30 °C to 1000 °C. To analyze the amount of hydrogen uptake for NiAl-LDH, the corresponding pulse tests also were carried out during the H_2_-TPR process. The outlet gas was connected with a drying bed filling with molecular sieve to absorb the moisture generated during the reduction process. H_2_-TPR experiments of NiAl-LDO and NiO/γ-Al_2_O_3_ were carried out with a similar process under the same amount of Ni content. For NiAl-LDO, about 0.1000 g of NiAl-LDH was loaded in a quartz reactor and then the precondition was changed to 500 °C for 4 h with continuous Ar flow to obtain NiAl-LDO. Furthermore, to ensure same amount of Ni content with NiAl-LDH, 0.2317 g of NiO/γ-Al_2_O_3_ (Ni 15.87 wt.%, ICP) was used for carrying out H_2_-TPR experiments and test conditions were the same as with NiAl-LDH. The H_2_ consumption signal of each sample in the whole H_2_-TPR process was monitored by a TCD detector, while H_2_ uptake was calculated using the following simplified equations.

Calibration Value:(1)$$CV=V_{s1}W_{H_{2}}/S_{\mathrm{pa}}$$

Volume:(2)$$V=CV\times S_{\mathrm{Sa}}$$

Total volume adsorbed:(3)$$Q_{\mathrm{tc}}=PV/(R(T+273.15))$$

Sample H_2_ consumption:(4)$$Q_{\mathrm{sc}}=Q_{\mathrm{tc}}/m$$
where *V*_s1_ is the loop volume of the sample cell (μL), *W*_H__2_ is percentage of analytical gas (H_2_, 10%), *S*_pa_ is the mean calibration pulse area, *CV* is the calibration value, *V* is the volume (μL), *S*_sa_ is the analytical area signals of TCD, *Q*_tc_ is the total volume adsorbed (μmol), *P* is the analytical pressure (atm), *R* is the standard coefficient (0.08206), *T* is the loop temperature (°C), *m* is the weight of sample (g), and *Q*_sc_ is H_2_ the consumption of sample (μmol g^−1^).

### 2.6. Catalytic Activity Evaluation

Catalytic performance tests of the catalysts were performed in a stainless-steel reaction column of internal diameter 8 mm. The superheated mixture of H_2_ and HDO (such as 100 °C, 150 °C, 200 °C, 250 °C, 300 °C, 400 °C or 500 °C) flowed to the reaction column with the same temperature. About 0.3150 g of NiAl-LDO (actual Ni 47.11 wt.%, ICP; theoretical one 58.57 wt.%) and about 0.9349 g of NiO/γ-Al_2_O_3_ (actual Ni 15.87 wt.%, ICP; theoretical one 17.19 wt.%) were used to ensure the same mass of Ni (14.84 wt.%) for the VPCE process, then samples were placed into a reaction column and heated up to reaction temperature in the flow of Ar. The total height of the catalyst layer was ~30–34 mm. The experimental apparatus was as outlined in Figure 1.

The H_2_ and vaporous HDO (molar feed ratio of deuterated water/hydrogen = 1, molar fraction of deuterium is 5.62%, molar fraction of hydrogen is 99.999%) were fed into the reaction column co-currently from the commingler to the reaction column. The HD concentrations in hydrogen gas at the reaction column outlet were analyzed with an Agilent 7890 gas chromatograph. The dedeuterium factor (*DF*) was applied for evaluating catalytic activity, defined as the following equation [29]:(5)$$DF=x_{t}/x_{b}$$
where *x*_t_ is the molar concentration ratio of initial deuterium in water, while *x*_b_ is the one in outlet water.

## 3. Results and Discussion

### 3.1. Material Characterization

A series of NiAl-LDH were prepared by adjusting synthesis conditions including hydrothermal temperature, crystallization time and Ni/Al/urea molar ratio, which were displayed in the Appendix A. The structural characteristics of the NiAl-LDH were evaluated from the XRD data, presented in Appendix A. In the powder XRD patterns of NiAl-LDH (treated by only adjusting hydrothermal temperature from 110 °C to 150 °C), a markable increase of (003) integral intensity were observed with increasing hydrothermal temperature, which indicated the rising content of the LDH crystalline phase in the NiAl-LDH. What’s more, the full width at half the maxima (FWHM) value of (003) reflections decreased in dependence on hydrothermal temperature, illustrating an increased crystallite size, in a good agreement with the calculated crystallite size (using Scherrer’s formula listed in Appendix A). The increasing crystallization time also resulted in the increases of (003) integral intensity and LDH crystallite size. However, the influence hydrothermal time on (003) integral intensity became unsharp from 12 h to 36 h, revealing that the effect of hydrothermal time on samples’ crystallinity was limited. In addition, the little effect of the Ni/Al/urea molar ratio was discovered with undramatic variation of XRD data (include integral intensity, half-height (W_1/2_), crystallite size) due to the similar alkaline environment provided by decomposing urea under a high hydrothermal temperature. Kovanda, F. et al. [30] and Chen, H. [31] reported similar changes in crystallinity with different hydrothermal treatments on properties of NiAl-LDH and microstructure control of oriented LDH films. The NiAl-LDH with the higher crystallinity in comparison with other samples were characterized by XRD analysis as shown in Figure 2A. The well-defined main diffraction peaks at 2θ = 11.7°, 23.5°, 35.1°, 39.7°, 47.3°, 61.2° and 62.5° correspond to the (003), (006), (012), (015), (018), (110), (113) crystal planes of NiAl-CO_3_-LDH (JCPDS No.15-0087), confirming the successfully preparation of NiAl-LDH [32]. After the calcining process, the disappearance of LDH characteristic diffraction peaks and the occurring of diffraction peaks at 37.1°, 43.4° and 63.2° typical for NiO (JCPDS NO. 01-1239) indicate the complete conversion of NiAl-LDH to NiAl-LDO [33]. At the same time, the absence of main peaks for Al_2_O_3_ in XRD pattern of NiAl-LDO (Figure 2A) is a demonstration for affirming the amorphous form of Al_2_O_3_ in NiAl-LDH and NiAl-LDO. In addition, as shown in Figure 2B, the XRD pattern of NiO/γ-Al_2_O_3_ demonstrate peaks of NiO and Al_2_O_3_, indicating the coexistence of NiO and Al_2_O_3_ phases (JCPDS 10-0425 [34]).

The morphology and micro-structure information of synthesized samples were investigated by SEM and TEM. As exhibited in Figure 3A, the prepared NiAl-LDH have an interconnected hierarchical structure and interlamellar petal-like nanosheets with a mean thickness of 18.7 nm. Its unique petal-like nanosheet was also reported in former literature [35,36]. Moreover, as shown in Figure 3B, specially superimposed sheets of NiAl-LDH could be observed, in good agreement with the above SEM results. Further EDS mapping of NiAl-LDH confirms the spatial distribution of chemical components (Figure 3B), demonstrating a homogeneous distribution of Ni, Al, C, O in NiAl-LDH. Inspiringly, even after calcination at 500 °C, the layered structure could maintain in NiAl-LDO (Figure 3C). Additionally, the TEM (Figure 3D) confirms the porous structure, which would promote ion diffusion and catalytic activity. What is more, the EDS mapping of NiAl-LDO shows the homogeneous distribution of Ni, Al, C, O, suggesting that the Ni species did not aggregate in the calcination process. The SAED patterns (Appendix A) exhibit well-defined diffraction spots, which are attributed to the (110) planes of the LDH and the (220) planes of the NiO. The findings are consistent with the XRD data (Figure 2A). These results indicate the monocrystalline nature of the NiAl-LDH and NiAl-LDO. However, Figure 3E shows that, in NiO/γ-Al_2_O_3_, a large amount of NiO aggregation on the surface of γ-Al_2_O_3_ during the processes of drying and calcination, which could block most of active sites and hinder the ion diffusion and charge transport. Further TEM of NiO/γ-Al_2_O_3_ illustrates the existence of aggregated NiO in γ-Al_2_O_3_ (Figure 3F), in good agreement with the above SEM results. Furthermore, the EDS mapping of NiO/γ-Al_2_O_3_ shows the heterogeneous distribution of Ni, Al in γ-Al_2_O_3_ (Figure 3F). The above results sufficiently prove the LDH-derived catalyst could maintain the layered structure and highly dispersed active species, which benefits the catalytic activity. Other LDH-derived catalysts have also been reported in other literature [36,37]. For instance, Li et al. [36] reported that the sulfureted NiAl-LDO could maintain a layered structure even after Na_2_S_2_O_3_ treatment and 400 °C calcination, exhibiting outstanding electrochemical activity.

To further affirm the porous structure in NiAl-LDO, low temperature nitrogen adsorption-desorption experiments were conducted. Comparing NiO/γ-Al_2_O_3_ with NiAl-LDH, as shown in Figure 4A,B, the much wider micropores of NiAl-LDO confirm the porous structure, which corresponds well to the TEM result (Figure 3D). The average pore size decrease (from 12.65 nm of NiA-LDH to 8.12 nm of NiAl-LDO, Figure 4D) further proves the formation of pores with a lower size. Moreover, the specific surface area of the calcinated sample increases more than two times when compared with LDH precursors (64.34 m^2^ g^−1^ of NiAl-LDH and 185.63 m^2^ g^−1^ of NiAl-LDO, Figure 4D). Besides, the specific surface area of NiO/γ-Al_2_O_3_ (173.43 m^2^ g^−1^) obtained by impregnating and calcining methods is approached with that of pure γ-Al_2_O_3_ (179 m^2^ g^−1^ [20]), but the dramatic declination of average pore size is discovered within it (18.1 nm [20] to 4.99 nm). The blocking of micro-pore caused by impregnating nickel in γ-Al_2_O_3_ surface could suppress the catalytic exchange on NiO/γ-Al_2_O_3_.

### 3.2. Catalytic Performance on Hydrogen Isotope Exchange

The VPCE catalytic performance evaluation apparatus was shown in Figure 1. As shown in Figure 5, the catalytic activity of the prepared NiAl-LDO is much higher than that of NiO/γ-Al_2_O_3_ under the same test condition from 200 °C to 400 °C, while NiO/γ-Al_2_O_3_ exhibits similar catalytic performance with NiAl-LDO at 500 °C. In industrial application, the VPCE process is preferred under a lower temperature with much fewer operating conditions and tritium permeation. Excitedly, catalytic activities of NiAl-LDO are more than nine times over that of impregnated NiO/γ-Al_2_O_3_ at 300 °C. Meanwhile, with the increment of temperature, the dedeuterium factor enhanced is obviously due to increased deuterium transfer from vapor to hydrogen. Besides, special structure characteristics (homogenous dispersion and flaky morphology, Figure 3C,D) of NiAl-LDO are available for the dynamic contact between gaseous water deuterium and hydrogen gas, leading to the higher catalytic activity of NiAl-LDO than NiO/γ-Al_2_O_3_ under a temperature lower than 400 °C.

XPS was used to evaluate the surface chemical states of catalysts before and after reaction. As shown in Figure 6A, the XPS spectrum of Al 2p at 73.71 eV is assigned to Al^3+^ and further proves the existence of Al_2_O_3_ crystallite in untested NiAl-LDO, which is in good agreement with the XRD results (Figure 2A). Meanwhile, the two spin-orbit doublets clearly locate at the binding energy of 855.8 eV and 873.4 eV, respectively, which correspond to Ni^2+^ 2p_3/2_ and Ni^2+^ 2p_1/2_. The accompanying shake-up satellites are located at 861.8 eV and 879.8 eV on the side of the Ni^2+^ 2p_3/2_ and 2p_1/2_ edge, respectively, illustrating the presence of Ni^2+^ in un-tested NiAl-LDO, in good accordance with the literature [36,38]. Comparing tested NiAl-LDO with an untested one, the slight shift of Al binding energy also is detected from 73.71 eV to 74.58 eV, which reveals stronger metal-support interactions between Al and Ni in tested NiAl-LDO. Besides, the tested NiAl-LDO exhibits Ni^0^ peaks, which are displayed as two weaker peaks at 852.4 eV and 870.1 eV, respectively [21,39]. Moreover, two spin-orbit doublets (located at 854.7 eV and 872.8 eV, respectively) and their shakeup satellites (shown at 859.6 eV and 878.9 eV, respectively) shift to a lower binding energy, demonstrating that the interaction of the Ni-O bond becomes weaker. Simultaneously, the relative intensity of the Ni^2+^ species decrease (from 100% of untested NiAl-LDO to 97.86% of the tested one, Appendix A), illustrating that the content of NiO decreases after the VPCE process. Comparing tested NiO/γ-Al_2_O_3_ with tested NiAl-LDO, as shown in Figure 6C,D, the XPS spectrum of Al 2p in tested NiO/γ-Al_2_O_3_ shifts toward a lower binding energy (from 73.66 eV to 73.60 eV). What is more, the two spin-orbit doublets (located at the binding energy of 872.83 eV and 855.21 eV, respectively) shift toward a higher binding energy (873.30 eV and 856.10 eV, respectively) after the reaction. These results confirm weak metal-support interactions between Al and Ni in impregnated NiO/γ-Al_2_O_3_. Meanwhile, the XPS spectrums of Ni^0^ in both tested catalysts were observed in Figure 6B,D, in good agreement with the corresponding XRD patterns (Appendix A). Above results reveals stronger metal-support interaction and both catalysts could be partially reduced under a test environment containing H_2_ and HDO. These two characters may be a benefit for higher catalytic activity, which would be discussed in the following section.

### 3.3. Relationship between Ni^0^ Species and Catalytic Activity

To further evaluate the influence of Ni^0^ species on catalytic activity, the catalytic performance of reduced NiAl-LDO (treated with different reduction temperatures under H_2_ atmosphere) were tested (Figure 7). After the reduction process, the catalytic activities of NiAl-LDO are obviously improved with the increase of reduction temperature below 300 °C. When reaction temperature > 300 °C, both catalysts (treated by a pre-reduction process at 500 °C and 700 °C, respectively) exhibit similar catalytic activities. Differently, compared with the un-reduced samples (NiAl-LDO), catalytic performances are observed for the VPCE process and *DF* exhibits an obvious promotion under the reaction temperature range of ~100–200 °C. The above experimental results demonstrate that the Ni^0^ species generated by an in-situ reduction of NiAl-LDO are able to promote catalytic activity below 300 °C for the VPCE reaction.

### 3.4. Reduction Property for Catalysts

To further characterize the reduction property and nature of metal-support interaction, the H_2_-TPR was conducted. As shown in Figure 8A, the TPR profiles of NiAl-LDH have a wide and strong peak centered around 598 °C and a shoulder peak at a lower temperature of 403 °C. The main peak is assigned to the reduction of well distributed Ni^2+^ species in the metal oxide, while the shoulder peak is attributed to the reduction of Ni^2+^ in the form of the bulk crystallite NiO species [40]. Comparing NiAl-LDO with NiAl-LDH, as shown in Figure 8B, the single reduction peak of NiAl-LDO is discovered at around 603 °C, which corresponds to the reduction of Ni^2+^ to Ni^0^. The higher reduction temperatures (than pure NiO around 370 °C [41]) of NiAl-LDH and NiAl-LDO could suggest stronger metal-support interactions between nickel and aluminum, which helps avoid nickel’s agglomeration in special test condition with H_2_ and then promotes the catalytic activity. The results also demonstrate that H_2_ has a good accessibility to reduce nickel completely for NiAl-LDH and NiAl-LDO. Importantly, the absence of the reductive peak of NiAl-LDO at <300 °C suggests that a reduction process cannot occur at below 300 °C and NiAl-LDO still maintains NiO components. Comparing the TPR profiles of NiO/γ-Al_2_O_3_ and NiAl-LDO, lower reductive peak (centered at 308 °C) indicates weaker metal-support interactions between Ni and γ-Al_2_O_3_, leading to NiO aggregation on the surface of γ-Al_2_O_3_ supports and lower catalytic activity in the reaction process. The two reductive peaks (located at 308 °C and 565 °C) are attributed to the reduction of Ni^2+^ interacting with near groups through the free Ni-O bonds [42] or the reduction of highly dispersed NiO species [41]. Meanwhile, it can be obviously seen that NiAl-LDH and NiAl-LDO have similar actual H_2_ consumptions (7429 μmol g^−1^ and 7086 μmol g^−1^, respectively) under the same loaded Ni, showing the similar reduction property, which corresponds well to the similarly centered reduction temperature (598 °C and 603 °C, respectively, Figure 8A,B). However, NiO/γ-Al_2_O_3_ consumes smaller H_2_ due to a different synthetical method. The actual H_2_ consumption of NiAl-LDH is lower than the theoretical one (9981 μmol g^−1^) and NiO/γ-Al_2_O_3_ catalysts show a similar phenomenon. On the other hand, catalytic activities of NiAl-LDO catalysts were observed at 250 °C in a reaction process while there is no reaction observed for NiO/γ-Al_2_O_3_. One of the reasons is the better Ni dispersion helps to enhance the reaction, another is that Ni^2+^ could bring an availably active site and benefit to catalytic activities.

The high temperature in the VPCE process prevents liquid water from blocking active sites of catalysts and therefore the whole transfer of deuterium occurs in a single stage, as shown in Equation (6):(6)$$\mathrm{HDO}(\mathrm{vapor})\;+\;H_{2}(\mathrm{gas})\;⇌\;\mathrm{HD}(\mathrm{gas})\;+\;H_{2}O(\mathrm{gas}),$$

As is well known, the catalytic reaction (Equation (6)) occurs on catalytic active sites. The catalytic reaction consists of the following steps: (1) enrichment of vaporous water and hydrogen; (2) diffusion and adsorption; (3) hydrogen isotope catalytic exchange; (4) desorption and diffusion. Based on the published research in science magazines, the pivotal mechanism is the dissociative chemisorption of D_2_O (or HDO) on Ni (111) crystal surface to form chemisorbed OD(ads), D(ads) or OH(ads), which has been confirmed by a quantum-dynamical investigation on a density functional theory (DFT) based global potential energy surface (PES) [43]. Meanwhile, hydrogen molecules can also dissociate into H(ads) on Ni (111) planes [44,45,46]. Therefore, it can be concluded that hydrogen and vaporous water are chemically adsorbed mainly on the Ni (111) planes and dissociate into a hydrogen atom, deuterium atom and hydrogen-deuterium atom. However, Figure 5 and Figure 8B show that catalytic activity exists at 250 °C and no reductive peak occurs at <300 °C for NiAl-LDO, suggesting that Ni^2+^ also bring catalytic performance in VPCE. In other words, another reaction pathway may coexist with the above-mentioned pathway. In the second pathway or route II, the dissociative chemisorption of H_2_ and HDO also occurs on the NiO species. Namely, there are two reaction pathways (route I and route II) for the VPCE on NiAl-LDO catalysts. Based on the above discussion, one possible pathway for the H-D isotopic exchange between hydrogen and water on metallic Ni and NiO can be deduced as follows:(7)$$H_{2}\;+\;2\sigma (M)\;⇌\;H(\mathrm{ads})\;+\;H(\mathrm{ads}),$$
(8)HDO + 2σ(M) ⇌ D(ads) + OH(ads),
(9)H(ads) + D(ads) ⇌ HDads + 2σ(M),
(10)$$H(\mathrm{ads})\;+\;\mathrm{OH}(\mathrm{ads})\;⇌\;H_{2}O\;+\;2\sigma (M),$$
where σ is the active site, while M represents Ni^0^ or NiO.

## 4. Conclusions

The vapor phase exchange reaction (VPCE) is a crucial important reaction for water detritiation and tritium extraction from heavy water. The high dispersed Ni-based catalyst derived from LDH is proposed and verified with better catalytic performance than traditional impregnated Ni/Al_2_O_3_ catalysts. Synthesis processes were systematically studied to optimize the preparation conditions and to improve the crystallinity of NiAl-LDH. As it was observed with the NiAl-LDH (synthesized at 150 °C for 12 h with Ni/Al/urea molar ratio of 2/1/14), a marked crystallinity was found by the adopted urea hydrolysis method. Moreover, after the calcinated process, in-situ grown Ni-catalysts derived from NiAl-LDH were applied firstly in the VPCE process between hydrogen and water. Experiments showed that NiAl-LDO catalysts exhibit an excellent reaction performance several times higher than that of impregnated NiO/γ-Al_2_O_3_ under a temperature of <400 °C, signifying a high potential large-scale application for VPCE. Then the better catalytic performance of NiAl-LDO was analyzed. Compared with impregnated NiO/γ-Al_2_O_3_, a unique petal-like nanosheet structure made the Ni-based catalyst (NiAl-LDO) quicker at ion diffusion and charge transport. Stronger metal-support interactions between Ni and Al_2_O_3_ supports prevented active metals’ aggregation. Furthermore, a homogenous dispersed metallic phase, higher specific surface area and wider pore with a smaller size provided more available active metals. These advantages led to a more excellent catalytic performance. Importantly, based on our discussions, we proposed the reaction pathways in the hydrogen isotope exchange between hydrogen and water. Both Ni^0^ species and NiO species showed catalytic activities in the VPCE process. This work highlights the application of an in-situ grown Ni-based catalyst derived from a NiAl-LDH precursor at a lower temperature in the VPCE process and the coexisting effects of NiO and Ni^0^ species on catalytic activities.

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
