# Peer review of "Ni-Based Catalyst Derived from NiAl Layered Double Hydroxide for Vapor Phase Catalytic Exchange between Hydrogen and Water"

_nanomaterials, 2019, doi:10.3390/nano9121688_

Round 1

Reviewer 1 Report

See attached file.

Author Response

We thank for your encouragement to our work and have already revised our manuscirpt based on your suggestions. Please download the attachment and see the attachment. We hope for your replies and see the responses to your comments or suggestion.

Reviewer 2 Report

The present manuscript reports on the synthesis, characterization and evaluation towards vapor phase catalytic exchange reaction of Ni-based catalysts derived by NiAl Layer Double Hydroxide (LDH). Although the reaction investigated is novel, various issues, shortly listed below, hinder the publication of the article in its present form:

The English language throughout the text should be carefully checked for typos/syntax errors. Please provide in the introduction section the latest advances in relation to the different catalytic materials used for VPCE process, in order the novelty and the necessity of the present work to be revealed. In the experimental section provide the nominal/theoretical Ni content (wt.%) for both catalysts. In the section 2.6 it is referred that the Ni content is 25.87% in Ni/Al2O3 whereas 47.11 in NiALLDO. Why different Ni content is used? Please explain why carbon element is present in as –prepared catalysts (Figure 3) Various arguments cannot be supported by the present results. For instance : “Above results sufficiently prove the LDH-derived catalyst could maintain the layered structure and highly dispersed active species, which benefits to the catalytic activity.”. Is the reaction under investigation structurally sensitive? Is the Ni dispersion the only factor that affects the reaction? This issues should be satisfactory addressed in order to support the above argument. Please correct “the specific surface area” instead “special surface area” Correct “185.63 m2g-1 of NiAl-LDO” instead of “185.63 m2g-1 of NiO/γ-Al2O3” Correct “binding energy” instead of “binging energy”. Provide the detailed reduction protocol followed (temperature, H2 concentration, time, temperature) for the experiments in Figure 7. In order the role of Ni reduced species to be disclosed a more thorough study/discussion is required. In particular, the high activity of Ni/Al2O3 at high temperatures cannot be justified by the present results. The XPS of Ni/Al2O3 before and after the reaction should be presented for comparison. Moreover, the impact of reduction pretreatment should be examined not only on NiAl-LDO but also to Ni/AL2O3 sample. Finally, the Authors have to take into account that XPS is an ex-situ technique not providing accurate information about the chemical state under the real reaction conditions. A re-oxidation of the sample is expected under XPS measurements. In summary, a thorough XPS study of all samples must be performed to support the present findings. The latter is of major importance towards obtaining reliable structure-performance correlations. At 500 C both catalysts exhibit similar performance, despite the very different Ni dispersion (as stated by Authors). This comes in contradiction to the argument that “highly dispersed active species, which benefits to the catalytic activity”. A more thorough discussion of the experimental findings is required. The H2 consumption in TPR tests (Fig. 8) should be compared to the theoretical one taking into account the NiO wt% in the three catalysts. This could provide information about the extent of reduction of NiOx species on the different catalysts. In this regard, the different catalytic performance can be possible explained by taking into account the easier reducibility of LDO samples under reactions conditions. Correct “strongly mental interactions” and “bigly specific surface area”. These terms are not convenient. The abstracts and conclusions sections should be re-written on the basis of the above comments. No evidences are provided in the present work about the role of Ni(111) facets on VPCE process, as refereed in the abstract.

Author Response

We thank for your encouragement to our work and have already revised our manuscirpt based on your suggestions. Please download the attachment and check the responses to your comments or suggestion.We hope for your replies

Round 2

Reviewer 2 Report

-